# Latin Transgender and Gender-Diverse Individuals’ Perceptions of Well-Being During the COVID-19 Pandemic [note 1]

**DOI:** 10.3390/bs14110997

**Published:** 2024-10-25

**Authors:** Hector J. Peguero, Dionne P. Stephens, Jacqueline Duong, Averill Obee

**Affiliations:** Department of Psychology, Florida International University, Miami, FL 33199, USA; stephens@fiu.edu (D.P.S.); jduong@fiu.edu (J.D.); aobee001@fiu.edu (A.O.)

**Keywords:** LatinX, transgender, gender-diverse, COVID-19

## Abstract

Latin transgender and gender-diverse (LTGGD) individuals experienced the public health measures mandated during the COVID-19 pandemic in unique ways. Intersecting ethnic and gender-identifying frameworks contributes to differing access, support, and well-being observations. The aim of this study was to examine emerging adult LTGGD individuals’ perceptions of their health and well-being experiences during the COVID-19 pandemic in South Florida. Interviews were conducted with nine self-identifying LTGGD individuals. Using a thematic analysis, a total of three major themes were identified as shaping the participants’ experiences and perceptions of health during COVID-19, including (a) healthcare interactions, (b) challenges, and (c) opportunities. Some of the participants were mixed in their perceptions of their well-being during the COVID-19 pandemic; negative concerns included the inability to access general and transgender/gender-diverse specific health services. However, others note that they were happy that the pandemic allowed them to avoid contexts where they regularly experienced microaggressions due to their identities. Additionally, various participants were able to use this time of isolation to identify providers with greater knowledge about LTGGD health needs. These findings highlight the importance of providing culturally competent and humility-centered skills to healthcare providers and others supporting the LTGGD population’s well-being.

## 1. Introduction

Although there was an explosion of research examining the COVID-19 pandemic’s impact on the perceptions of individual health access and support-informed outcomes, cross-cultural, LatinX transgender, and gender-diverse (LTGGD) emerging adults were largely overlooked. This is concerning, given that LTGGD individuals identities converge three vulnerable risk groups—emerging adults, LatinX, and gender and sexual minorities. Studies have shown that COVID-19’s negative impact on well-being is much higher among emerging adults when compared to other age groups [1,2,3]. LatinX people make up 33% to 37% of people in the United States who were reported as COVID-19 cases, including fatality rates [4,5]. Similarly, COVID-19 had a significant impact on sexual and gender minorities’ well-being. Transgender and gender-diverse (TGGD) individuals experienced worsened mental health, reduced access to basic resources, economic instability, inconsistent social support, and substance use outcomes during this time [6,7,8].

Few studies have focused specifically on the experiences of the LTGGD population during COVID-19 [9]. This gap in research is particularly concerning, given the high levels of risk they face. The current study seeks to fill this knowledge void by examining how LTGGD emerging adults perceived healthcare during the pandemic. Understanding these factors is crucial for comprehending how LTGGD individuals navigated their health experiences during this period. This information is essential for identifying potential leverage points for health interventions targeting this population in future health crises and in general health practice.

### 1.1. Theoretical Frameworks

This study uses an integrated module to illustrate LTGGD individuals’ perceptions of health and well-being during the COVID-19 pandemic. Bronfenbrenner’s Socioecological Theory [10,11] provides the foundational framing of the connections across environmental, contextual, and social systems. Building upon this, intersectionality illustrates the ways in which LTGGD individuals’ identities inform their power positionalities within these systems.

#### 1.1.1. Bronfenbrenner’s Socioecological Theory

Bronfenbrenner’s [10,11] ecological perspective suggests that various environmental factors influence our development when looking at human development and how individuals interconnect with environmental systems. This theory emphasizes that our interactions with the people, institutions, and communities around us shape our growth and behavior. Developmental contexts include family, peers, schools, neighborhoods, communities, and culture/society, as well as the impact of historical events. Bronfenbrenner also suggests that ecological theory is bidirectional, where the individual shapes their immediate environment, known as the microsystem [10,11].

Bronfenbrenner’s model has identified multiple levels of systems that individuals negotiate. These include the mesosystem, as defined by the ecological system framework, referring to the interconnected relationships and the involvement of emerging adults within different settings [10]. These settings include the home where an individual lives, schools, non-profit organizations, and other individuals with varying degrees of power. The exosystems, on the other hand, refer to areas where the emerging adult is not yet involved but can still impact their development. Finally, the macrosystem encompasses anything related to the individual’s cultural beliefs and influences that may affect their transition to adulthood. The ecological framework allows one to focus on the interrelationships within their environment and the mutual influences that exist between them.

Studies examining sexually diverse LatinX individuals, for example, have noted that experiences with healthcare systems are often impacted by their sense of support from familial and community influences [12,13,14]. Concerns about healthcare access and historical experiences with healthcare systems were also found to influence their outcomes, which are examples of the macrosystem, mesosystem, and exosystem levels in the Bronfenbrenner model.

#### 1.1.2. Intersectionality

It is important to consider which individuals have multiple identities when examining health outcomes. Furthermore, the ways in which the powers/privileges associated with these identities make individuals vulnerable to health inequalities cannot be ignored. For this purpose, we incorporated an intersectionality lens. This paradigm asserts that individuals’ multiple social identities together shape their experiences and their positions of power across systems [15]. Intersectionality provides a framework, looking at how, socially, historically, and politically, many factors can influence people’s identities, especially those with multiple identities interacting and influencing one’s health experiences and well-being [15,16,17]. Intersectionality underlines the importance of how power, privilege, and oppression lead to people’s daily life experiences. Thus, beyond looking at individuals as members of singularly defined groups (e.g., gender, sexuality, race/ethnicity), intersectional approaches allow for a more critical lens through which to observe the differing positions and changing powers afforded to individuals across changing contexts or moments [18,19]. This involves ensuring that examinations of these phenomena in research addresses (1) who is included within this category, (2) what role inequality plays, and (3) where there are similarities [15].

### 1.2. COVID-19 Pandemic

The COVID-19 pandemic from 2019 to 2021 had consequences across all groups in the United States of America, including health interactions (e.g., unavailable mental health resources, poor access to treatments, loss of insurance, etc.), challenges (i.e., return to the parental home, social interactions, discrimination, etc.), and opportunities (i.e., social distancing, education delays, etc.) [1,9,20,21]. These realities were found to have differing levels of impact across populations, however, as those from marginalized populations reported greater negative outcomes [22]. Of particular importance were the vulnerabilities that racial/ethnic and gender/sexual minority populations experienced.

Studies have noted that Latinos perceive COVID-19 as having negatively impacted themselves, their loved ones, and the larger community in many ways [23,24,25]. Along with the disproportionate rates of acquisition, hospitalization, and death, many also reported experiencing greater financial and social stressors during the pandemic [9,21,26,27]. Familial members, church, and community health centers were identified as important sources of support during this time [9,25,28,29].

Although there is little research specifically on LTGGD emerging adults, research examining lesbian, gay, bisexual, queer, asexual, and intersexual (LGTBQAI+) individuals have noted that several factors influenced their perceptions of wellness during the COVID-19 pandemic. These include sexual and gender minorities experiencing a decrease in access to supportive resources, such as organizations that offer specific health services [30]. They also faced a lack of support from family members or others who may not have accepted their identity [9,31,32,33,34]. In addition, sexual minorities faced challenges in accessing gender-affirming medical care, which created unique experiences for them [35]. Again, these factors are not unique to the COVID-19 period, as many TGGD individuals experience physical, emotional, and mental harm both before and after the COVID-19 pandemic because of societal norms and structural inequalities [36]. For example, heterocisnormativity, the assumption that heterosexuality and cisgender identity are the norm, is pervasive in healthcare settings [29,37]. The lack of knowledge about LTGGD general health concerns became even more acute during COVID-19, when health service access became limited and less personalized.

#### Current Study

This study seeks to identify the challenges that LTGGD emerging adults negotiated during the COVID-19 pandemic. The use of the Bronfenbrenner model guides our identification of factors that shaped LTGGD individuals’ perceptions of their well-being during COVID-19. To better understand their power positionality and social location as marginalized individuals negotiating these systems, the integration of an intersectional lens was used.

## 2. Methods

For this study, a qualitative approach was used to explore LTGGD individuals’ socially constructed perceptions of their well-being during the COVID-19 pandemic. The approach taken is useful for addressing sensitive issues faced by marginalized and stigmatized communities [38]. The use of one-on-one interviews was employed to ensure confidentiality and provide opportunities for thick and rich data collection [39].

Interview and survey data were collected via Zoom interviews between September and December 2020; a county-wide state of emergency was implemented and canceled halfway through this interview period. Initial lockdown practices included physical distancing requirements, travel restrictions, governmental work-from-home orders, the closure of non-essential businesses and some workplaces, and the restriction of health services to mainly virtual consultations until early October. However, all the interviews were conducted via Zoom to ensure health and confidentiality protocols were consistent throughout the data collection period. Ethical approval for this study was received through the authors’ university’s Institutional Review Board (IRB).

### 2.1. Participants

A total of nine participants completed a survey and interview, ensuring that we reached information and data saturation [40,41,42]. The survey collected demographic identity data (see Table A1 and Table A2 in the Appendix B and Appendix C). Demographics were gathered at the beginning of the study, and pseudonyms were assigned to each participant for confidentiality throughout the interview, transcription, and data analysis processes.

The participants identified as follows: Queer (*n* = 4), Straight (*n* = 1), Polyfluid (*n* = 1), Lesbian (*n* = 1), Bisexual (*n* = 1), and Pansexual (*n* = 1). Regarding gender identity, three identified as Non-Binary (*n* = 3), one as a Transgender Man (*n* = 1), two as Trans Masculine (*n* = 2), one as Non-Binary Gender Fluid (*n* = 1), one as Gender-Fluid Intersex (*n* = 1), and one as a Transgender Woman (*n* = 1). All the participants identified as LatinX, with familial nationalities of origin including Mexican (*n* = 1), Cuban (*n* = 4), Puerto Rican (*n* = 2), Puerto Rican/Mexican (*n* = 1), and Colombian (*n* = 1). All the participants provided informed consent, as they were all 18 years of age or older.

### 2.2. Measures

The participants first completed an online survey that gathered demographic information and their chosen pseudonym/nickname for use in the study. The interview guide was developed using an ecological approach. Questions focused on individual and system-level factors related to their perceptions of well-being during COVID-19. (An edited version of the interview guide is provided in Appendix A).

### 2.3. Recruitment

After receiving the approval of the IRB at the authors’ university, where all the researchers were based, the recruitment was done through communication with LGBTQIA+ community organizations, The Health Department of Miami Dade in Florida, and ads (flyers) placed on Instagram, Facebook, and other social media platforms. The ads had information about the study and compensation information, detailing that they would receive a $20 Walmart gift card for their time. Most of the interviews were scheduled after a first contact by email, asking participants to participate in the study. Then, the one-on-one interview was scheduled for approximately 3–5 days after the first communication.

### 2.4. Data Collection

The participants completed both the online survey and a semi-structured interview about their health experiences during the pandemic. One hour prior to the meeting, the participants were sent the IRB-approved consent form to review and a Zoom link. To further ensure the participants’ confidentiality, directions for how to keep the audio on and camera off in Zoom were also included in this email message. Once the participant logged into Zoom, the first author again went through the consent and ensured that it was understood. All the participants consented to participate.

The online survey was hosted on the Qualtrics platform, an institutional-supported online survey program. The first survey question required the participants to enter a pseudonym/nickname to protect their identities and serve as the participant identification code. The survey took an average of 10 min to complete. Once they were finished, the the participants let the interviewer know they were ready to begin the interview.

The first author (HP) conducted all the interviews via Zoom. The interviewer first requested the participant to share the pseudonyms/nicknames they entered in the survey; this was used to refer to the participant throughout the interview. The interview guide focused on Bronfenbrenner’s level of analysis; this meant that the questions explored the perception of individual (individual level), familial, ethnic communities (microsystem level), and social systems’ influence (mesosystem and exosystem level; see Appendix D). The ways in which the participants’ diverse identities shaped these responses were guided by the intersectionality approaches noted by Cole [15].

To ensure the information shared by the participants during the interviews was comprehensive, the interviewer utilized prompt questions to summarize and confirm their responses. Throughout the process, the interviewers were careful to avoid leading questions and refrained from making assumptions about how the participants’ experiences were shaped by the pandemic. Each interview lasted approximately 45–60 min. After the interview, the participants received an informational booklet with contacts for community support counseling services, should they wish to discuss their experiences further. The audio recordings from Zoom were promptly downloaded and transcribed, with the transcripts being checked and rechecked by three different research assistants.

### 2.5. Data Analysis

After collecting the data, the researchers analyzed the information using thematic analyses, following the six-step guidelines by Braun and Clarke [43] and utilizing Excel. The authors independently reviewed the transcriptions provided by the research assistants, identifying key points and commonalities within the data. Based on their collective findings, the first and second authors developed an initial codebook (see Appendix B), which was refined through ongoing discussions among the authors as coding progressed across the transcripts.

The codebook included categories such as “Health Physical”, “Health Mental”, “Loss of Social Support”, and “Uncertainty”. Each author independently applied codes to the transcripts, but the final codes were determined collaboratively. Once the final codes were established, the themes and categories were re-examined in relation to each individual interview. For each text segment, codes were created based on the problems presented within the themes. Additionally, new codes were introduced as emerging themes were identified. Any discrepancies or disagreements were addressed during weekly team discussions until a consensus was reached.

### 2.6. Credibility and Trustworthiness

To ensure credibility and trustworthiness, several strategies were employed, including the use of open-ended questions, probes, and interviewer notes [44,45]. Validity checks were performed by detailed reviews of the transcripts and audios by research assistants who were not involved in the data collection and through a review of the interviewers’ and the transcription checkers’ notes. To address confirmability concerns, debriefing sessions were conducted throughout the data collection and analysis phases to explore preconceptions and increase reflexivity practices [46].

### 2.7. Positionalities

The first author identifies as a member of the LGBTQIA+ (Lesbian, Gay, Bisexual, Transgender, Queer, Intersex, Asexual, and others) community; he is a gay, cisgender man of Puerto Rican descent. Being an “insider” in this way has motivated his engagement in research examining diverse sexual and gender minorities globally, with an underlying goal of increasing the knowledge and insights that those who do not identify as LGBTQIA+ may not have [47].

## 3. Results

The focus on COVID-19-specific experiences provided unique insights into the ways in which healthcare and well-being experiences across the ecological systems’ three levels were informed by identity-level factors. A total of three broad themes were identified from the data as being relevant to understanding LTGGD individuals’ health and well-being perceptions during the COVID-19 pandemic in South Florida: (1) healthcare interactions, (2) challenges, and (3) barriers. We present the data within the themes as they align with each layer of the systems described in the ecological model. The challenges and opportunity themes were applicable across differing ecological levels, namely the microsystem, mesosystem, and exosystem. In contrast, the theme of healthcare is primarily clustered under the individual and exosystem levels.

### 3.1. THEME 1: Healthcare Providers’ LTGGD Knowledge

Healthcare providers’ knowledge was defined as the education and training necessary to address LTGGD-specific health concerns and needs. This knowledge emerged as the most critical factor influencing LTGGD emerging adults’ perceptions of their health and well-being. The participants viewed healthcare providers as essential sources of information, support, and resources related to their health needs. Within this theme, three subthemes were identified as particularly important: (1) general LTGGD healthcare knowledge, (2) LTGGD-specific mental health knowledge, and (3) access to LTGGD-specific health resources.

#### 3.1.1. General LTGGD Healthcare Knowledge

Three out of nine of the participants (33.0%) noted that the general knowledge of medical providers was a primary reason for the lack of services and understanding of the needs of LTGGD emerging adults. One participant emphasized the need for improved sex education, particularly regarding reproductive health within the medical field. This need was especially relevant for individuals who are transitioning and were designated female at birth. In other words, it is crucial for healthcare providers to possess this knowledge to alleviate the burden on care-seekers, eliminating the necessity for patients to educate these professionals about transgender health needs.

“*I suppose uh there’s a lot of, I mean in general, there’s very little [reproductive organs] education. But I think that things of the reproductive organs in the medical field really need to step up, especially for people who are designated female at birth*”.(*Nathan*)

This knowledge was perceived as important, not just for physical well-being but also for their perceptions of engaging with providers. One participant, for example, noted that their physician’s explanations about a unique medical service focusing on hormone therapy were so critical, not only to ensure that they would avoid negative medical reactions but also for building trust in their physician.

#### 3.1.2. LTGGD-Specific Health Resources Access

It emerged that the ability to be connected to organizations and clinics that were knowledgeable about the LTGGD community was critically important. Healthcare providers were perceived as being important sources of information for this, via referrals and knowledge about existing agencies.

“*I’m very fortunate in that I attend [local university] and that [local university] gives me access to free mental health services so I’m able to be part of groups on campus and things like that*”.(*Dalia*)

“*So there’s actually resources that provide trans affirming doctors, and it has a sort of master list, and so if you go to a physician and have a negative experience you can actually tell them and this doctor will be added to a list, you know don’t go to this doctor, or if you have a very positive experience, and a very good doctor that helps, and is helping affirm your identity, instead of shutting you down, then you can also report that too, and it will be added to a list of trans affirming doctors, which is very beneficial to the community, because we don’t want to have to go through those negative, oops, I just popped off my headphone, one second, we don’t want to go through those negative experiences ourselves and so getting advice from people who have already been to those physicians, can sometimes, can be the difference between life and death*”.(*Jinx*)

#### 3.1.3. LTGGD-Specific Mental Health Knowledge

The findings highlighted the significance of mental health resources and the impact of mental health providers on individuals’ well-being. Culturally competent mental health services were identified as a primary need, particularly for many LTGGD individuals who navigate micro and macroaggressions on a daily basis. One participant underscored the importance of well-meaning mental healthcare providers, emphasizing that these providers should be knowledgeable about the specific mental health concerns faced by LTGGD populations. This participant pointed out that medical providers can inadvertently create barriers to well-being by lacking essential knowledge about the community and their mental health needs.

“*It’s very difficult to find well-meaning mental health clinician or well-meaning doctors. But if they’re not educated around nonbinary and trans gender identities, they can do a lot of harm. And so, I think that’s one of the biggest barriers to well-being is people just not having the education or awareness that they need to in order to really serve the community*”.(*Dani*)

### 3.2. THEME 2: Challenges

When asked about health challenges experienced during the COVID-19 pandemic, familial relationships were the only source mentioned. This was particularly salient among five of the participants, who were residing with their families at the time of their interview. A total of three of the participants had been in college prior to the pandemic shutdown and were forced to move back home. It is also important to note that family as a source of challenges differed in the ways it affected perceptions of health. Specifically, families’ influences were discussed in terms of affecting day-to-day interactions, rather than broader healthcare access or care issues. These responses emerged as critical around two subthemes: (1) familial tensions and (2) familial responses to mental health.

#### 3.2.1. Familial Tensions

Approximately 40% (*n* = 4) of the LTGGD emerging adults lived with their parents, facing stress-related problems in family relationships at some point. In comparison, heterosexual emerging adults do not face many of the challenges that LTGGD individuals face in relation to sexual and gender topics [48]. Because of the pandemic isolation and having to go back to their family homes, the participants describe elevated negative interactions with family members. Some of the participants expressed that negative interactions were related to the increased time they were spending together with their families and the stress and conflicts that had arisen since the beginning of the COVID-19 pandemic.

“*Oh, it’s actually been on both sides of the spectrum. So, on one side, having to be back in my, being kicked off campus and having to go back home in the environment with my family which is not always the best for me was very difficult*”.(*Jinx*)

“*COVID-19 completely ruined my plans and it’s been very, very hard to cope with it… like emotionally, mentally, spiritually and also like I get into fights a lot with like my family as well and it is what it is*”.(*Mari*)

#### 3.2.2. Familial Response to Mental Health

The participants also shared how family members were not supportive of mental health topics. This was particularly acute when considering concerns the participants had related to their LTGGD identities. One participant reported how difficult it was to continue attending therapy to support their LTGGD needs during the pandemic due to familial interference.

“*As a trans person going to therapy is very important just so you can talk about your feelings (?) gender and be able to just know what you’re going through. Having to do that while also being Hispanic is hard. When I first wanted to go to therapy it was incredibly difficult to convince my parents to let me go. Because first they were like “Well what are we doing?” They thought it was because of them. Second of all they were like “Well why can’t you pray about it?” And so that is not something that can be done with depression and anxiety*”.(*Ang*)

### 3.3. THEME 3: Opportunities

Although the above participants reported negative perceptions about their health experiences during the pandemic, there were also several opportunities found to support their well-being positively. An overarching benefit was being able to avoid the microaggressions that they experienced when seeking support for their health and well-being.

“*Yes so, yeah so, having access to transforming doctors, you know, a lot of the times you go to the doctors and it’s very difficult because a lot of doctors still see trans as a mental illness. Having a doctor that thinks that is delusional. Instead of actually supporting your identity and helping you get to where you wanna be- is a very big concern*”.(*Jinx*)

More specifically, the participants note that the COVID-19 pandemic gave them time to (1) engage in mindful self-reflection and (2) avoid misgendering. Both subthemes highlight individual-level mental well-being benefits of the isolation required due to the COVID-19 pandemic.

#### 3.3.1. Mindful Self Reflection

Several of the participants (*n* = 3) described aspects of their lives as experiencing a moment during the pandemic of getting to know themselves. Living through the pandemic allowed these individuals to have time to acknowledge the importance of interactions with family and friends, as well as acceptance of who they are as a person.

“*The good part about it is it really helps when you have so much time with yourself you really learn yourself, you listen to yourself, you have time to be with your family and of course it is hard because you guys are with each other all the time*”.(*Raye Raye*)

“*Um it has allowed me to understand myself better and grow to accept myself for who I am*”.(*Nathan*)

#### 3.3.2. Misgendering

Experiences of misgendering—the use of wrong pronunciation or words that do not accurately define their gender—were viewed as stigmatizing and distressing. Several of the participants (*n* = 5) became aware of this and more appreciative of the COVID-19 pandemic isolation because it allowed them to avoid situations where this would occur.

“*This might sound weird. I’ve mentioned like one of the biggest barriers for me is like being misgendered but being that I’m hardly in public that hasn’t really happened. So, I think that’s been really good*”.(*Dani*)

“*Instead of having to deal with a lot of social situations, in which I was constantly being misgendered, I could just be in my room or surrounded by people who are socially distancing but comfortable with and who understands my… my identity*”.(*Nathan*)

## 4. Discussion

One of the key findings of this study was the general lack of satisfaction the participants had with their providers’ general LTGGD healthcare knowledge. During the interviews, the participants noted that many physicians had little information to address their health concerns beyond transitioning; further, those who were designated female at birth were more likely to report that their healthcare provider lacked knowledge about their specific health needs.

### 4.1. LTGGD-Specific Mental Health Knowledge

While many factors can impact an individual’s health, this study found that the LTGGD community continues to face barriers that prevent medical providers from understanding their lifestyles and behaviors, which are related to their primary health needs at the exosystem level. The participants reported negative experiences that stemmed from stigma related to sexual orientation and gender diversity, as well as barriers to accessing mental health care due to systemic discrimination within the healthcare system—a finding supported by the existing LTGGD literature [49].

Moreover, many individuals experienced stigma being initiated by medical providers, along with a low level of understanding regarding how gender and sexually diverse identities shape health outcomes. This lack of understanding was directly linked to negative interactions within mental health services [50,51]. The participants emphasized the importance of promoting equity and diversity in mental healthcare systems, particularly regarding gender and sexual minority statuses.

Additionally, they highlighted the significance of acknowledging how their LatinX identity intersects with their experiences, as this was central to their cultural and health values. Increasing healthcare providers’ social skills and knowledge specific to gender and sexual diversity is essential for enhancing LTGGD individuals’ engagement with and adherence to mental health support systems [52].

### 4.2. LTGGD-Specific Health Resources Provision

It is essential to mention the importance of healthcare resources needed for LTGGD individuals. Access to ordinary healthcare was emphasized in the study because of financial encounters, a lack of insurance access, and education about specific medical services that LTGGD individuals need. Similarly, other LatinX, heterosexual, low-income communities in the United States of America also faced financial encounters and the lack of resources related to the healthcare system in the USA [9,53].

Many of the study participants noted that unique medical services, like hormone therapy, most of the time were not correctly performed because access was not found, or they did not trust their medical providers to perform this type of treatment. These unique transgender-related medical needs encounter structural barriers because of the lack of education about hormonal therapy by medical providers. This exosystem concern reflects the need for the improved training of health care providers [54]. However, macrosystems were also relevant as attitudes and trends in health insurance were critical. The refusal of this type of treatment by insurance companies, who do not want to cover the cost of hormonal therapy and even more surgical protocols and procedures, aligns with prior research showing that these cultural pressures impact mental well-being [14,55]. Our findings demonstrate the importance of an approach needed by the LTGGD community, focusing on gaps and improving their health. One of the approaches most needed is the training of healthcare professionals in relation to gender identity, up-to-date transgender healthcare medicine, and the prohibition of discrimination and the denial of services [54,56].

## 5. Challenges

### 5.1. Familial Tensions

Even before the COVID-19 pandemic started, research showed that the exosystem and macrosystem levels are critical for marginalized and stigmatized individuals’ well-being. This was reflected in the current study, as the participants reported the need for better education in the healthcare system by medical providers to avoid negative reactions and to be able to trust physicians. Several of the participants had challenges regarding their mental health because of the lack of support by their families in this regard, creating stigmas around mental health services. This aligns with prior research, noting that LatinX families may have difficulty addressing the sexual and gender diversity issues experienced by sexual minority individuals [51,57]. Further, the participants in the study felt at peace when not having to experience misgendering by others, making it feel stigmatizing and stressful. In another study, TGGD individuals experiencing familial conflicts due to LatinX cultural and religious values reported general distress [6]. These factors contribute to increased health disparities and adverse health outcomes in the LTGGD community, becoming stronger when family rejection is visible during isolation times [20]. Like for four of the participants, isolation became a space where these individuals felt uncomfortable showing who they really are as a person. They also were uncomfortable with the increased time they had with family members, mainly because of the lack of understanding about their sexual identity.

### 5.2. Familial Responses to Mental Health

Moreover, besides having to fight about their identity and uncomfortable moments with their families, the participants also had to face the lack of education LatinX families can have about mental health. Many mentioned the lack of support that LatinX families showed when it came to mental health issues, particularly in relation to their LTGGD identity. Most commonly, family members continued to interrupt the participant’s mental health therapy because they did not believe in mental health treatments or viewed religion as a more appropriate way to address mental health [54]. This supports Moyce et al., [58], who found that the participants in their study expressed that they preferred to find other ways to maintain their mental health by concentrating on their emotions, physical, and spiritual health. Studies have shown that family connectedness is critical for gender and sexual minority individuals’ mental health, especially because of societal prejudice, stigmas, and discrimination [59]. This is even more important because when it is compared to other types of support, such as peer support, family support helps shape LGBTQIA+ people’s mental health outcomes, like, for example, anxiety and depression [9,60]. For LatinX populations, this becomes even more pronounced, given that prior research has noted that family support is closely tied to health outcomes and well-being for transgender LatinX individuals [14,61].

## 6. Opportunities

### 6.1. Mindful Self Reflection

The findings also noted that the pandemic provided some positive mental health space. Specifically, the participants shared that during this period, they had more opportunities to get away from negative experiences outside their homes. They reported that being able to avoid these microsystem and mesosystem sources of stress increased their sense of positive well-being. This supports prior research conducted with this community, wherein, hostility, insensitivity, and the poor ability to treat patients have been reported to create barriers to these individuals’ physical and mental health [62,63]. For example, a study that focused on LTGGD individuals found 45.7% of patients in the emergency department witnessed medical staff mocking, gossiping, and making jokes about these individuals, and 62.9% had experienced healthcare professionals purposefully engaging in misgendering [63]. These negative experiences contributed to the participants mistrust of health providers and the broader healthcare system [63,64].

Having the opportunity to stay home and to not deal with microaggression allowed several of the participants to engage in mindfulness and self-reflection. Mindfulness is defined as a non-judgmental awareness that allows an individual to explore the ways that we think about ourselves and others [36]. People who experience significant vulnerability due to traumas are able to cultivate self-compassion and coping skills through this practice; these participants reported that this was a key focus during their time at home during the pandemic. This individual-level practice has been found to be an accessible and positive coping skill cross culturally [65]. This is particularly important for marginalized populations who often do not have access to services or training to address mental health concerns [66].

### 6.2. Misgendering

Lastly, in the study, several of the participants mentioned the topic of misgendering, particularly at mesosystem levels. The participants experienced misgendering when in their daily lives, which was viewed as stigma and moments of distress. These feelings also led to feelings of gender dysphoria due to misgendering. It is mainly seen when looking at gender-diverse individuals’ studies that show that they can face more misgendering when it comes to the use of correct pronouns [67]. This specific form of microaggression discounts a person’s personal reality, which can lead to poor levels of identity coherence and strength [67,68]. However, in the study, the participants also mentioned that due to having to deal with situations of misgendering in their everyday lives, they appreciated COVID-19 and the related isolation resulting in them not having to deal with this situation.

## 7. Limitations

This study is critical for expanding the literature on LTGGD individuals; however, it has important demographic and study design limitations that should be considered. A significant demographic limitation is the lack of a more diverse sample of LatinX TGGD individuals in the community. Specifically, there is considerable diversity across gender, sexual, and ethnic identities, and the nine participants in this study cannot fully represent the wide range of experiences within their respective groups [42]. Future research should aim to gather a larger, more diverse sample to build on the findings presented here.

Additionally, it is important to consider that the participants’ experiences may have been influenced by their geographic location. The region where the data were collected has a large LatinX population and resources available for TGGD individuals. In contrast, LTGGD individuals living in other areas of the country may face different responses to their health needs due to lower population levels. Furthermore, the participants’ responses are context-specific, as each state approached the COVID-19 pandemic differently.

In the state where this study took place, two intersecting factors played a significant role. First, the government and related systems were among the earliest in the nation to reopen during the pandemic. Consequently, many healthcare services became overwhelmed with both COVID-19-related demands and existing health concerns. This led to the decreased prioritization of and access to specific medical services that are essential for LGBTQ individuals, such as hormonal therapy [69,70]. Additionally, political policies in the state limited access to services for transgender individuals, resulting in the deferral of gender-affirming treatments during this period [25].

Study design limitations include the impact of self-selection, which can lead to a self-selection bias. This bias results in unequal representation, as certain groups may be less likely to participate [71]. In this study, this could mean that only individuals who felt strongly about the topic chose to participate, limiting the perspectives represented and excluding others within the transgender and gender-diverse community who might have different experiences.

Additionally, the use of interviews as a research method may have influenced some of the participants to withhold certain information, opting instead to provide responses that they believed were culturally appropriate or socially desirable. Conducting interviews exclusively in English may have further restricted the participants from using specific terminologies or expressions with nuanced meanings. Consequently, Spanish speakers might have felt more comfortable sharing their perspectives in Spanish, which could better convey their personal viewpoints.

## 8. Conclusions

Despite some limitations, this study provides essential foundational knowledge about the perceptions and well-being experiences of LatinX transgender and gender-diverse individuals during the COVID-19 pandemic. It also highlights how gender identity and ethnicity intersected with the supports available to them during this unique period.

The findings indicate that a lack of education among healthcare providers negatively impacted the participants’ sense of well-being during the isolation of the pandemic. The participants expressed that the general healthcare system was not equipped to address their specific needs, with misgendering being a common issue that further exacerbated their experiences. These adverse healthcare outcomes underscore the importance of healthcare providers understanding the dynamics between gender identity, sexual orientation, and access to care.

Additionally, relationships between the participants and their family members were strained during the pandemic. The participants reported a lack of understanding from their families regarding their mental health needs, largely due to insufficient education about gender and sexual identity. For many, moving back home created tension within the household, interfering with their mental, physical, and emotional health.

Interestingly, the COVID-19 pandemic offered individuals the opportunity to practice self-care and engage in mindful self-reflection. This self-reflection allowed the participants to gain a deeper understanding of themselves and their future aspirations. As a result, they could explore new ways of thinking about themselves and others, free from the microaggressions that had been prevalent before the pandemic. This opportunity also enabled individuals to cultivate self-compassion, which was particularly important given the daily traumas many faced.

These findings highlight the importance of comprehending the relationships between gender, sexuality, race/ethnicity, and welfare, particularly in different contexts, healthcare, and cultures. Our study serves as a starting point for transforming the conventional understanding of the health frameworks for sexual and gender identities by broadening the view of the challenges faced by minority populations in terms of their specific well-being needs. Although the study focused on the perception of the well-being of LTGGD individuals, acknowledging this will enhance efforts to address overall well-being issues among minority populations, especially LTGGD individuals.

## Data Availability

The datasets presented in this article are not readily available because the data are part of an ongoing data analysis. Requests to access the datasets should be directed to [hpegu003@fiu.edu].

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
