# Peer review of "Latin Transgender and Gender-Diverse Individuals’ Perceptions of Well-Being During the COVID-19 Pandemic†"

_behavsci, 2024, doi:10.3390/bs14110997_

Round 1

Reviewer 1 Report

Comments and Suggestions for Authors

THe findings presented do not clearly show differences between the study participants (as representatives of the LTGGD emerging adult population) and other non-Latinx, non-LTGGD people in the same age group during the COVID pandemic. Other Latinx youth having had to move back home in the pandemic experienced similar issues with religious families not being open to the youth getting mental health services. Other non-Latinx youth from religious families too. THe paper does not make a strong case for having findings that are unique to the intersectionality of the group of focus. 

Comments on the Quality of English Language

The paper needs to be read carefully as some sentences are not clear or are not complete. I have highlighted them in yellow in the attached file.

Author Response

Thank you very much for taking the time to review this manuscript. Please find the detailed responses below and the corresponding revisions in track changes in the re-submitted files. All highlighted areas have been addressed by adjusting incomplete sentences. Some of the sentences are correct and are terminology used in social sciences. The extra spaces funded in the document were mistakes from transferring the article to the journal format. The questions related to emerging adults LTGGD individuals in comparison to heterosexual emerging adults have been addressed. You will find all changes in the document attached to this portal.

If you have any more questions, do not hesitate to contact me.

Have a great week!

Hector J. Peguero 

Reviewer 2 Report

Comments and Suggestions for Authors

The study is pioneering in the field of non-conventional gender identification and more precisely in conjunction with the intersectionality of socioeconomic status, ethnicity, race and gender. Researching and exposing oneself in such an - unfortunately, still - slippery area carries with it inadvertently risks, one of them being that the final presentation of the research in the form of as scientific paper must appear as close to flawless as possible. The study needs some editing as well as some proof-reading. Please see attached file for some more concrete suggestions towards improvement.

Comments on the Quality of English Language

The study needs proof-reading and editing.

Author Response

Thank you very much for taking the time to review this manuscript. Please find the detailed responses below and the corresponding revisions in track changes in the re-submitted files. Moreover, I have corrected the grammar in some of the sentences that you have pointed out in the reviewed document.  We have also added more information in the study limitations s you have suggested. 

If you have any questions, do not hesitate to contact us.

Have a good week1

Hector J. Peguero

Round 2

Reviewer 1 Report

Comments and Suggestions for Authors

Thanks for addressing my earlier comments. Please read carefully the additions in particular, because not all sentences are complete (e.g. missing verb)

Comments on the Quality of English Language

It's not an issue with the English language, but rather with needing to be read again to make sure all sentences are complete.

Author Response

Thank you very much for taking the time to review this manuscript. Please find the detailed responses below and the corresponding revisions in track changes and, highlighted in the re-submitted files. Some sentences have been slightly changed and corrected. You will find all changes in the document attached to this portal.

If you have any more questions, do not hesitate to contact me.

Have a great week!

Hector J. Peguero